# 3D Model Artificial Intelligence-Guided Automatic Augmented Reality Images during Robotic Partial Nephrectomy

**DOI:** 10.3390/diagnostics13223454

**Published:** 2023-11-16

**Authors:** Michele Sica, Pietro Piazzolla, Daniele Amparore, Paolo Verri, Sabrina De Cillis, Federico Piramide, Gabriele Volpi, Alberto Piana, Michele Di Dio, Stefano Alba, Cecilia Gatti, Mariano Burgio, Giovanni Busacca, Angelo Giordano, Cristian Fiori, Francesco Porpiglia, Enrico Checcucci

**Affiliations:** 1Department of Oncology, Division of Urology, University of Turin, San Luigi Gonzaga Hospital, 10043 Orbassano, Italy; danieleamparore@hotmail.it (D.A.); paoloverri05@gmail.com (P.V.); sabrinatitti.decillis@gmail.com (S.D.C.); federico.piramide@gmail.com (F.P.); alb.piana@gmail.com (A.P.); marianoburgio@gmail.com (M.B.); giovanni.busacca95@gmail.com (G.B.); a.giordy@gmail.com (A.G.); cristian.fiori@unito.it (C.F.); porpiglia@libero.it (F.P.); checcu.e@hotmail.it (E.C.); 2Department of Surgery, Candiolo Cancer Institute, FPO-IRCCS, 10060 Candiolo, Italy; pietro.piazzolla@gmail.com (P.P.); volpi_gabriele@yahoo.it (G.V.); gatti.cecilia@libero.it (C.G.); 3Division of Urology, Department of Surgery, SS Annunziata Hospital, 87100 Cosenza, Italy; micheledidio@yahoo.it; 4Romolo Hospital, 88821 Rocca di Neto, Italy; stefanoalba78@gmail.com

**Keywords:** artificial intelligence, 3D models, robotics, kidney cancer, augmented reality

## Abstract

More than ever, precision surgery is making its way into modern surgery for functional organ preservation. This is possible mainly due to the increasing number of technologies available, including 3D models, virtual reality, augmented reality, and artificial intelligence. Intraoperative surgical navigation represents an interesting application of these technologies, allowing to understand in detail the surgical anatomy, planning a patient-tailored approach. Automatic superimposition comes into this context to optimally perform surgery as accurately as possible. Through a dedicated software (the first version) called iKidney, it is possible to superimpose the images using 3D models and live endoscopic images during partial nephrectomy, targeting the renal mass only. The patient is 31 years old with a 28 mm totally endophytic right-sided renal mass, with a PADUA score of 9. Thanks to the automatic superimposition and selective clamping, an enucleoresection of the renal mass alone was performed with no major postoperative complication (i.e., Clavien–Dindo < 2). iKidney-guided partial nephrectomy is safe, feasible, and yields excellent results in terms of organ preservation and functional outcomes. Further validation studies are needed to improve the prototype software, particularly to improve the rotational axes and avoid human help. Furthermore, it is important to reduce the costs associated with these technologies to increase its use in smaller hospitals.

## 1. Introduction

Today, we are in the era of precision medicine, which is increasingly being applied in the surgical field, including the treatment of oncological pathology [1]. To achieve this goal, surgery has begun to take advantage of three-dimensionality for the study of the patient’s anatomy, pursuing the creation of a surgery procedure completely tailored to the patient [2]. In addition, this technology can aid the physician during the preoperative counseling phase, discussing with the patient about the anatomy and the phases of the intervention. In fact, although it may be considered a minor part, informing the patient and counseling about the procedure that will be performed is a key step. Recognizing the organ, understanding the type of surgery and the approach used allows the patient to have more confidence and peace of mind than traditional counselling in which the patient feels a sense of missing out.

In this manner, it is possible to better understand the information obtained from two-dimensional instrumental investigations (i.e., contrast-enhanced Computed Tomography (CT) and Magnetic Resonance Imaging (MRI)), currently still representing the pre-operative standard for surgical planning. In addition, 3D models can be used also for training and simulation, thanks to their high level of anatomical details [3,4]. In fact, virtual 3D models’ use during partial nephrectomy (PN) aids the surgeon to perform a safe procedure and represents a consistent predictor of PN success, its use being associated with a two-times higher probability of achieving trifecta, regardless of the different definitions available in the literature [5]. This is because 3D reconstructions allow the surgeon to preoperatively study the clinical case more precisely and to mentally plan a more suitable surgical approach and a precise planning. This instrument aids the surgeon during the renal pedicle’s management, during both the identification and management phases, performing a clamping, pushing the indication of a selective approach.

Furthermore, planning an accurate resection strategy is directly associated with a more effective collecting system preservation and a precise medullary and cortical reconstruction, decreasing postoperative complications rate. Beyond the demolitive and enucleative part of the procedure, it is essential to have a proper reconstructive phase as well. Very often, this phase, also within the fact that the renal artery is clamped, is performed in a concise manner to try to minimize ischemia time that may impact residual function. Having precise planning of this phase of the procedure allows for more precise suturing in less time and is thus reflected in greater residual renal function.

The building in the mind reconstruction process is not immediate, but it requires some mental training in order to gradually become easily appliable in the day-to-day practice [6]. With the aim to overcome this issue, new software for 3D reconstruction were developed and have been increasingly available for physicians. Thanks to these tools, a clear visualization of the anatomy in 3D resulted in simplifying the understanding of clinical cases by both surgeon and patient. Several published studies have shown how such a technology can play an essential role in preoperative planning in all settings to pursue the path of precision surgery [7,8,9,10]. Detailed understanding of surgical anatomy is the key point for a tailored treatment planning, especially when performing renal surgery and renal function preservation, which currently plays a key role [11,12,13]. Consider, for example, patients who have already impaired renal function before surgery (e.g., chronic renal failure from diabetic or hypertensive nephropathy); patients who are functionally single kidney; or patients who are surgically single kidney. Preservation of renal function must be ensured even more in these patients, and therefore the role of three-dimensional reconstructions and precision surgery have even greater impacts.

Expanding the indication of such a technology, namely, intraoperative surgical navigation with the superimposition of the 3D models and performing an augmented reality procedure, is gaining increasing popularity. The accuracy of the 3D overlapping together with the usefulness of AR visualization during kidney sparing surgeries has already been proven. Its field of application is extremely heterogeneous and can range from renal superimposition to allow a more precise enucleation and increased parenchymal sparing to the intraoperative superimposition to identify atheromatic plaques during renal transplantation [14] or to perform a correct renal puncture (of the lower calyces) during percutaneous procedures for stone treatment [15].

In the first case, for example, the use of 3D AR has enabled surgeons to overcome one of the main limitations of renal transplantation with a living-donor robotic approach by laying the groundwork for extending the indication of this procedures to patients with advanced atheromasic vascular disease. However, regardless of the technique, the main limitation of this technology lies in the necessity to receive an adequate training needed to correctly superimpose the virtual models on the real intraoperative field. In the latter case, on the other hand, studies have moved towards the direction of augmented reality because, despite technical and technological innovations, percutaneous puncture is still the most challenging step in performing percutaneous nephrolithotomy. This maneuver is, in fact, characterized by the steepest learning curve, with consequent increased risk of injury to both the punctured kidney and the surrounding organs. In this experience, in all the cases, the inferior calyx was correctly punctured in accordance with preoperative planning, using the superimposed hologram, with reduced procedural time and significantly less radiation exposure time. Finally, no increased intraoperative or postoperative complication rates were recorded.

So far, however, the superimposition process is limited by the need to perform the overlapping manually: given the peculiarity of this process, it must be performed by an expert operator, who must be present throughout the procedure. Furthermore, due to the need for a dedicated team (i.e., engineers and urologists) familiar with this platform, this technology is currently only available in few specialized centers, and its use is therefore not yet popular. One of the latest innovations in this field was developed in order to overcome this limit and is represented by the creation of a dedicated software that can automatically superimpose 3D virtual models on the real anatomy to perform automatic augmented reality procedures [16]. In a previous study, we already had demonstrated how automatic superimposition during partial nephrectomy is a feasible process thanks to the use of indocyanine green (ICG) and a specially developed software, the first version, called IGNITE (Indocyanine GreeN automatIc augmenTed rEality), which allowed for superimposing the 3D reconstruction and the intraoperative images of the real organ by a using computer vision approach in an average time of 7 s, allowing precise and safe enucleoresection of endophytic renal masses [17]. However, the main limitation of this technology was related to intraoperative deformations of the kidney, which resulted in altered anatomy, resulting in inaccurate automatic anchoring. This has been and still is the main problem to overcome, making it necessary to implement organ recognition using, for example, deep learning algorithms.

Therefore, at the state of the art, there are currently no studies related to the use of artificial intelligence in automated guidance for augmented reality partial nephrectomy. This study is a pioneering work in the field and an opener for future studies on the role of artificial intelligence in this field.

In this case report, we present the first case of robotic partial nephrectomy with artificial intelligence-driven automatic superposition.

## 2. Case Presentation

### 2.1. Software Descriptions

In this Section, we present the main technical aspects of our software solution, called “iKidney”, in the following, and used it to overlay the 3D model of the patient’s kidney to its real counterpart framed by the endoscope, during in vivo RAPN. The goal of iKidney is to determine the six degrees of freedom of the real kidney with respect to the endoscope camera framing it. To achieve the desired result, the software leverages a Convolutional Neural Network (CNN) as the tool for processing the images received in real-time from the endoscope, inferring from them the rotation and position of the kidney.

We use transfer learning on the pre-trained ResNet.50 model with a dataset composed of tagged images of kidneys that we assembled. We selected ResNet, short for Residual Network, because it has shown exceptional performance in image recognition tasks, winning first place in the ImageNet Large Scale Visual Recognition Challenge (ILSVRC) in 2015 and 2016. ResNet can learn extremely complex features and representations, which makes it ideal for image-based tasks.

This ResNet model’s main purpose is to segment the endoscope image in real-time, creating a mask that contains those pixels belonging to the kidney parenchyma only. Once this mask is composed, the organ’s rotation and position can be determined by analyzing the geometric properties of the segmented kidney. These properties include the location of the kidney’s center of mass, the orientation of its major and minor axis, and the dimensions of the kidney. The analysis is performed using a combination of computer vision algorithms and ad hoc devised heuristics. During the surgical procedure, the degrees of rotation available along the Cartesian axis to the kidney are limited because of the presence of the blood vases entering the kidney and anchoring it to the rest of the patient’s body. This allows the software to exclude from the analysis results some of the rotations.

Once the position and rotation of the kidney has been determined, the iKidney solution is able to use them to correctly overlay the 3D kidney model over the endoscope video image and to stream the augmentation back to the robot console, to assist the surgeon at request into the Tile-pro window.

The tests performed during in vivo RAPN proved the iKidney solution to be effectively able to infer the rotation and position of the kidney, resulting in a visually coherent overlapping, as assessed by medical experts, as long as at least one of the axes of the kidney mask is completely visible. Future works will include a formal study about the software’s precision using appropriate metrics. The main challenge is to create a precise correspondence between the 3D model axes origin and the anchor point used to overlay the model to the endoscope image. As there is no automatic method available yet, minimal human assistance is required for the initial registration in terms of a rotation and/or translation correction. In future releases of the software, we will address this limitation to decrease the time costs for the adjustment.

### 2.2. Clinical Case

The case in object is a 31 years old male patient. His past medical history included laparoscopic appendectomy. During an abdominal ultrasound performed to further study persistent stypsis, a right renal mass at the upper pole 28 mm was detected. The lesion was further studied with a contrast-enhanced CT scan, which confirmed the present of a solid nodule with a PADUA (Preoperative Aspects and Dimensions Used for an Anatomical) score of 9, and a R.E.N.A.L (Radius, Exophytic/endophytic, Nearness of tumor deepest portion to the collecting system or sinus, Anterior/posterior, Location relative to the polar line) score of 7a. After collegial discussion, the patient was admitted to our department to perform a robotic assisted partial nephrectomy.

Consent for surgery and the use of 3D technology with automatic superimposition was obtained for the patient. The total operative time was 98 min with a selective clamping time 23 min. Total blood loss was 420 cc, not requiring any intra- or postoperative blood transfusion. The hemoglobin value on admission was 15.4 g/dL, and the hemoglobin value on discharge was 13.3 g/dL. The length of hospital stay was 3 days; a bladder catheter was intraoperatively positioned and subsequently removed on second postoperative day (POD). Histological examination revealed a clear cell renal cell carcinoma (ccRCC), stage pT1a, with negative surgical margins (R0). The patient had no in-house and 30-day postoperative complications according to Clavien–Dindo classification.

### 2.3. Intraoperative Navigation

Intraoperatively, 3D virtual models can be viewed and consulted during surgery on demand by the first surgeon in a cognitive manner, with the aim to maximize the benefits of the real-time navigation technology. After trocar placement, the renal lodge and the main anatomic landmarks (i.e., principal renal artery, renal vein, collateral arterial and venous branches, renal pelvis and ureter) were identified, being constantly helped by the 3D reconstruction during the identification process: the renal artery divided itself into a prepyelical and a retropyelic branch (i.e., a secondary branch). The prepyelical branch was composed of three secondary arteries: one apical, one superior, and one inferior branch. To perform a selective clamping, an empirical strategy based on the direction of arterial branches can be chosen, with the risk of failures during partial nephrectomy, even when assisted by three-dimensional virtual models. Therefore, as already shown in a previous work [18], we have developed a mathematical algorithm, aiding the surgeon to plan the best strategy to perform an effective selective clamping.

This algorithm is based on the Voronoi diagram and is capable of calculating the Euclidean distance thanks to dedicated mathematical formulas, therefore estimating the perfusion areas of the renal parenchyma from the single arterial branch. These areas are therefore displayed on the kidney with different colors, creating the so called “rainbow-kidney”, which allows the surgeon to accurately plan a selective clamping strategy, avoiding potential vascular damage (i.e., devascularization and revascularization) to the healthy kidney tissue. In addition, this minimizes the risk associated with the performance of selective or super selective clamping, which could lead to uncontrolled and mismanagement of the renal mass if wrongly executed.

Thanks to our “rainbow kidney” model, we divided the kidney into seven parts, each one vascularized by a specific artery, and so the first operator could decide which one to clamp to perform a selective clamping. During the procedures, the apical and superior artery of the prepyelical artery was performed to revascularize the superior pole and preserve the remaining renal parenchyma. At this point, the automatic overlay software was activated to recognize the location of the totally endophytic renal mass (Figure 1). Subsequently, following the 3D model’s projection on the organ, the lesion’s contour was marked on the renal parenchyma’s surface. Then, using a laparoscopic ultrasound probe, the precise location of the renal mass was confirmed (Figure 2). After performing selective ischemia, indocyanine green (ICG) was injected to verify the effectiveness of the selective ischemia strategy, showing a total devascularization of the upper pole (Figure 3), as predicted. An enucleoresection was performed followed by a reconstruction of the medullary and renal cortical parenchyma (Figure 4).

## 3. Discussion

As suggested by a recent study published by Porpiglia et al. [19], the use of 3D virtual models as an assessment tool for intraoperative surgical navigation provides the surgeon with a more accurate understanding of the details of a renal mass. This instrument allows, together with nephrometry and surgical complexity, which can help to predict intra- and postoperative complications, for a better understanding of the complexity of a renal lesion [20]. The use of this technology confirmed the advantages associated with 3D-guided surgery, as already highlighted in previous published works [7,21,22,23,24]. In particular, the surgeon can preoperatively study the case after consulting the model. The relationships between the tumor lesion, the vascular and urinary systems were preoperatively analyzed and subsequently confirmed intraoperatively thanks to our software, being able to visualize the model within the robotic console using the Tile-pro included in the DaVinci console. The possibility of having a better understanding of the renal vascular tree with a precise representation of the perfusion areas could represent a tremendous breakthrough in the management of selective clamping, as the concept of segmental vasculature would be largely reinterpreted leading to a better intraoperative management of selective clamping and excellent sparing of renal function [18].

Furthermore, these models can be also intraoperatively utilized with an on-demand consultation (i.e., cognitive manner) or with a real-time superimposition with the real anatomy (i.e., augmented reality procedure). The already demonstrated accuracy of AR during the overlapping phase, and its usefulness during intraoperative guidance is limited by a manual overlapping performed by surgeons’ assistant. In fact, to perform the overlap, an expert assistant must be present during the procedure since the surgeon cannot do it by themselves. Furthermore, the manual process may indeed lead to potential mistakes during the superimposition due to different factors such as the interpersonal differences in the perception of the operatory field, the different level of expertise of the assistant, the misunderstanding of kidney rotation, and more.

Therefore, the need to automatize the overlapping process represents a priority, which has been explored with preliminary attempts using computer vision algorithms.

This is the reason why we specifically developed artificial intelligence-based software with the aim to obtain an accurate automatic overlapping. Automatic superimposition makes it possible to overcome those limitations inherent in manual superimposition and, more importantly, allows the surgeon to be able to perform this procedure independently, without the necessary presence of another expert surgeon.

Even if our proposed solution is not devoid of limitations, such as the sporadic lack of preciseness in anchoring the model to the actual kidney, the tests performed during in vivo RAPN showed that the iKidney software can indeed infer the rotation and position of the kidney, resulting in a visually consistent (as assessed by medical experts) overlay, provided that at least one of the axes of the kidney mask is fully visible. However, during the training phase of our software, based on 15 prerecorded surgeries with more than 15,000 photograms for surgery and more than 225,000 for all surgeries, anchoring precision has been assessed using the standard IoU metric (Intersection over Union) metric [25], comparing the overlapping of the bounding box of the segmented kidney with that of the superimposed 3D model, returning an average overlapping precision of 97.8%.

Future work will include a formal study of software accuracy using appropriate metrics. The main challenge is to create software able to accurate match the origin of the axes of the 3D model with the anchor point used to superimpose the model over the endoscopic images. As an automatic method is not yet available, minimal human assistance is still required for the initial registration (i.e., fixing the rotation and/or correcting the translation). In future versions of the software, we will address this limitation to reduce the initial adjustment time, or better, to eliminate it. In addition, another major limitation is certainly the cost that this technology requires; in fact, not all centers may have software or hardware permitting to automatically or non-automatically superimpose a three-dimensional kidney model on intraoperative images; however, the advantages, particularly in selected cases, such as young patients, patients affected by chronic kidney problems in whom the preservation of kidney tissue is fundamental, and patients with a single kidney, are clear and tangible. We are currently in an embryonic phase of the project, and therefore the costs are still high. One of the objectives is to allow an ever-increasing number of centers to use this technology for the evident advantages and to reduce costs. If this technology became the property of every center that performs oncological surgery or in any case precision and saving surgery, the cost of using it would be reduced.

Finally, when we consider our iKidney software, which relies on advanced AI algorithms, we encounter important legal and ethical concerns. These issues encompass defining roles and responsibilities within the decision-making process, addressing privacy concerns regarding patient data used in training the network and ensuring rigorous quality control over the software’s performance.

## 4. Conclusions

In conclusion, we believe that this technology is fascinating for the urological world and promising for the future of surgery, but validation studies will be needed in order to use it in all renal oncology surgery settings, and, evidently, studies performed on large numbers will be needed in order to define the net benefit that such a technology could provide. The more data that are provided to the artificial intelligence software, the more accurate the automatic overlay determining the variabilities can be overcome by the software’s experience.

## Figures and Tables

**Figure 1 diagnostics-13-03454-f001:**
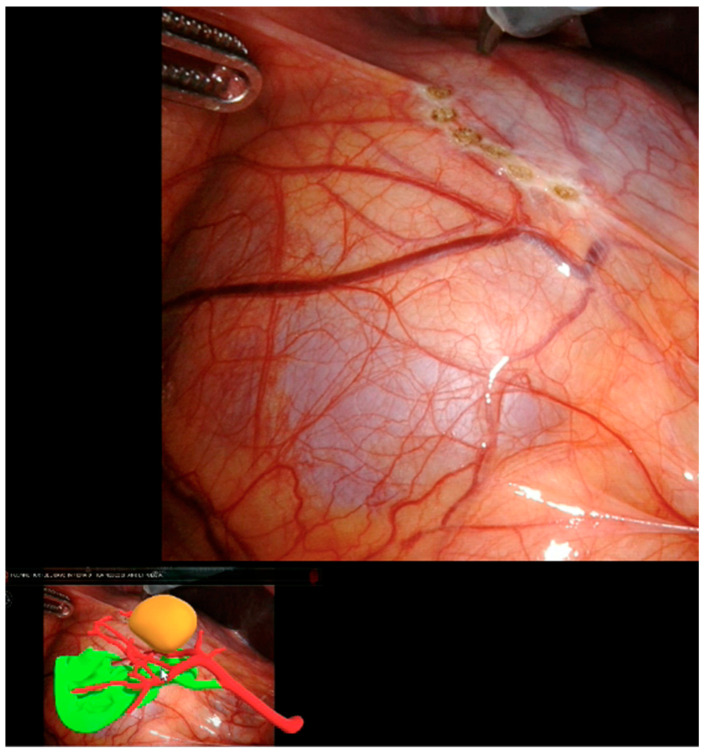
Automatic kidney overlay and identification of kidney mass projection.

**Figure 2 diagnostics-13-03454-f002:**
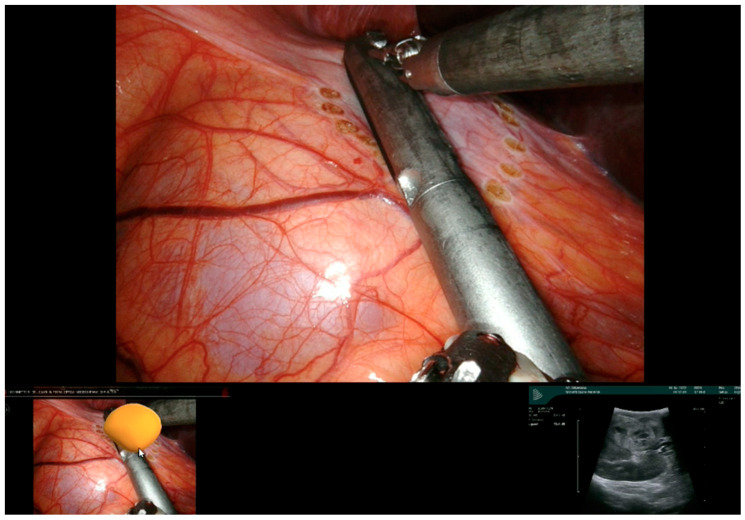
Confirmation by ultrasonography of renal parenchyma marked on the surface.

**Figure 3 diagnostics-13-03454-f003:**
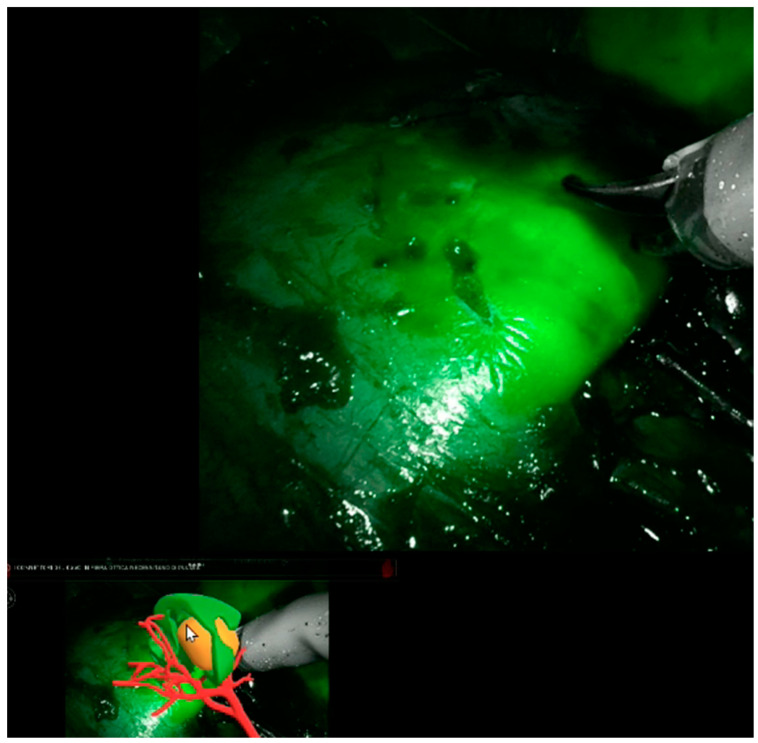
Devascularization of the upper renal pole.

**Figure 4 diagnostics-13-03454-f004:**
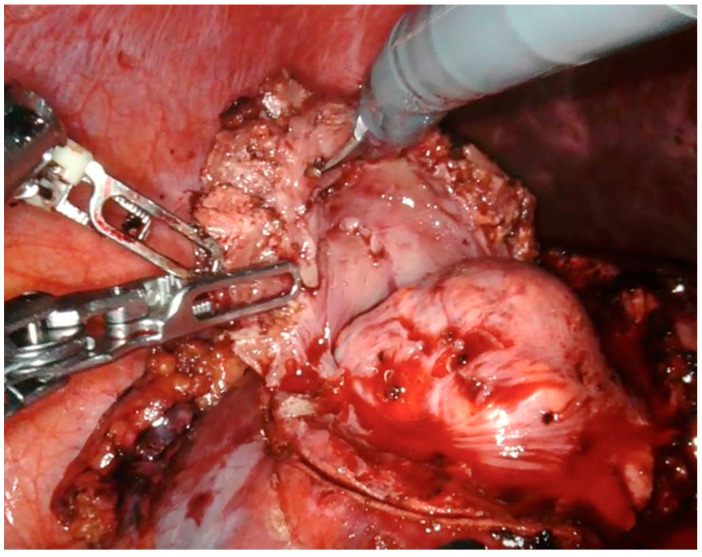
Enucleoresection of endophytic renal mass.

## Data Availability

Data are not available as this is a case report.

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
