# Peer review of "3D Model Artificial Intelligence-Guided Automatic Augmented Reality Images during Robotic Partial Nephrectomy"

_diagnostics, 2023, doi:10.3390/diagnostics13223454_

Round 1
Reviewer 1 Report (Previous Reviewer 4)
Comments and Suggestions for Authors
The author has answered my questions well and clarified some technical issues. This version of the article has met the standards for magazine publication, and it is recommended to accept publication.
Reviewer 2 Report (Previous Reviewer 1)
Comments and Suggestions for Authors
Thanks for your corrections.
This manuscript is a resubmission of an earlier submission. The following is a list of the peer review reports and author responses from that submission.
Round 1
Reviewer 1 Report
Comments and Suggestions for Authors
Please more describe the following items:
-The novelty aspects of your work in comparison with recent state-of-art.
-Framework or system diagram of your proposed AI assisted 3D AR model.
-The limitations of AI-assisted model and error rate of your proposed method.
-Performance comparison of proposed method with traditional partial nephrectomy in some tables or charts.
-Please check for any possible typos.
Author Response
Dear Reviewer, we have made the requested corrections that can be found in the text highlighted in green. As for typos, they have been corrected and highlighted in yellow. On the other hand, regarding the third and fourth points where it is requested to highlight the limitations of the model and the error rate and comparison with traditional partial nephrectomy, it is not possible now to evaluate these data. In fact, this is still an embryonic project and the cases treated so far are too few to compare with the traditional method. The future goal is to enroll patients in a randomized trial to evaluate surgical outcomes understood as oncologic and functional outcomes.
Reviewer 2 Report
Comments and Suggestions for Authors
Dear Authors,
This report, states the role of the precision medicine application in the surgical field, in the context of treating oncologic pathology. The 3D technologies are used to study patients' anatomy and create personalised surgical procedures and for training and simulation, surgeons can improve surgical outcomes and patients.
The significance of 3D reconstitutions is preoperative planning, especially in renal surgeries and function preservation. Patients with impaired renal function require careful planning to ensure functional conservation.
I suggest this report be accepted for publication as it shows the importance of a non-invasive method application to increase the efficiency of renal surgeries.
Author Response
Thank you very much for your comments and professionalism.
Reviewer 3 Report
Comments and Suggestions for Authors
well written good described on 3D Model Artificial intelligence guided automatic augmented 2 reality images during robotic partial nephrectomy. Results are valuable for fyrther studies.
Author Response

(The authors gave the same response as above.)

Reviewer 4 Report
Comments and Suggestions for Authors
This article introduced a method of 3D model artificial intelligence guided automatic augmented reality images during robotic partial nephrectomy. The idea is interesting, however, I have two questions.
The practical use and feasibility of this method need to be discussed, as it is very unnecessary for surgeons with proficient surgical skills. Qualified trained doctors can evaluate the morphology of tumors through preoperative image reading. For completely endogenic kidney tumors, the main difficulty lies in localization, but intraoperative ultrasound can effectively solve this problem. For the vast majority of kidney tumors, there is a tumor capsule, and after accurate localization, resection becomes relatively simple. I can't imagine that this intraoperative real-time 3D software model is of great practical significance to a urologist who has mastered robot surgery skills. Perhaps it is more significant to train junior doctors.
It has certain intraoperative auxiliary significance for assisting in the dissection and occlusion of arterial branches during surgery. But for real surgery, general partial nephrectomy can dissect the primary branch of the renal artery in the renal hilum. However, the article divides the kidney into 7 parts, each of which can be precisely blocked by arterial branches. I cannot understand that if the 7 parts are not the primary branch of the renal hilum area, they are branch blood vessels that exist within the kidney and cannot be blocked.
Author Response
Dear Reviewer, the practical use of 3D virtual models with AR and particularly as highlighted in our study with automatic superposition is a valuable aid especially in complex renal masses. In fact, the future goal is to extend the use of this technology to those masses that would otherwise be treated by radical nephrectomy. In addition, the advantage of this technology over ultrasound is constant and perennial guidance during both the resection and reconstruction phases to better highlight calyceal and vascular structures.
Regarding the second question, in our center we constantly use super selective clamping of even order II or III branches. In fact, our studies have used the Voronoi mathematical model. From an artery model visualization of the arterial phase of a computed tomography scan, a center line is extracted and represented as a set of seed points laid on consecutive bidimensional planes to adapt the 3DVM to the algorithm. Each kidney parenchyma voxel is associated with a center-line point (ie, a seed) according to a proximity criterion using the mathematical Euclidian distance from a seed point of the arterial center line to define each perfusion area. The final product computed by the algorithm with all the data for each bidimensional plane and its relative center-line points and areas, which are combined consecutively to create a 3DVM of the parenchyma with distinct regions of vascularization, each rendered in a specific color. Obviously, in cases of difficult dissection of arterial branches, clamping of a higher order is performed. But having the possibility of preserving a portion of parenchyma that is irrigated by a branch that is not clamped certainly has advantages in terms of functional outcomes.
Round 2
Reviewer 1 Report
Comments and Suggestions for Authors
The authors claim that it is not possible to evaluate the error rate of the proposed method and compare it with traditional methods. So, it is hard for the reviewer to justify the novelty and significance of the proposed method.
Reviewer 4 Report
Comments and Suggestions for Authors
The author has answered my questions well and clarified some technical issues. This technology has certain prospects. After revising the article according to the requirements of other reviewers and the article can be considered for acceptance.